# Mechanisms of Copper Toxicity and Tolerance in the Aquatic Moss *Taxiphyllum barbieri*

**DOI:** 10.3390/plants12203607

**Published:** 2023-10-18

**Authors:** Martin Bačkor, Michal Goga, Pragya Singh, Viktória Tuptová

**Affiliations:** 1Department of Biochemistry and Biotechnology, Institute of Biotechnology, Faculty of Biotechnology and Food Sciences, Slovak University of Agriculture, Tr. A. Hlinku 2, 949 76 Nitra, Slovakia; 2Department of Botany, Institute of Biology and Ecology, Faculty of Science, Šafárik University, Mánesova 23, 041 67 Košice, Slovakia; michal.goga@upjs.sk (M.G.); pragya.singh@student.upjs.sk (P.S.); viktoria.tuptova@gmail.com (V.T.)

**Keywords:** amino acids, antioxidants, copper, glutathione, metals, mosses, phenolic acids, pollution

## Abstract

Aquatic habitats are very frequently polluted with different kinds of xenobiotics, including heavy metals. For biomonitoring studies of aquatic pollution, algae are frequently used, as they do not contain protective cuticle on the surface of their thalli and can accumulate pollutants over the whole surface of thalli. However, this is a feature of most cryptogams. For this reason, we assessed the sensitivity of the aquatic moss *Taxiphyllum barbieri* (Java moss) to copper excess in a short-term study. Moss *T. barbieri* belongs to the common aquatic plants originating from Southeast Asia. To test the sensitivity (or tolerance) of the moss to excess Cu, selected concentrations (50, 250 and 500 µM) were employed in our 24 h studies. Total and intracellular Cu accumulation positively correlated with Cu availability in the water. This total and intracellular Cu accumulation was negatively correlated with decreased intracellular K content. Excess Cu negatively affected the composition of assimilation pigments and soluble proteins. Cu caused increased peroxidation of membrane lipids assessed using TBARS assay. Excess Cu decreased GSH to GSSG ratio and ascorbic acid content. We did not observe phytochelatin synthesis in this moss. The roles of selected amino acids, their intermediates and derivatives, as well as S-containing nucleosides and phenolic acids in Cu homeostasis and toxicity or tolerance were evaluated. We assume that this moss has potential for future employment in water quality evaluation.

## 1. Introduction

The rapid increase in population in the past few years and the problems related to different kinds of pollution and contamination of natural resources have become important subjects for discussion. The impact is not only limited to the death and extinction of small organisms but also affects the whole planet, including the variety of flora and fauna. As we know, water resources are requirements of every living organism, and that is the reason why they need to be studied keenly. There is a wide range of pollutants possibly responsible for water pollution. Among all of the types of pollutants, heavy metals are one of the most dangerous pollutants [1,2].

Irrespective of their size, shape and ecotypes, plants have played a vital role in the study of the primary factors causing various changes in the ecosystem. It is also helpful to recognize the types of elements present in the environment resulting in contamination of water. Many recent studies have demonstrated the use of lower plants as biological indicators to investigate the situation. The most abundantly growing life form in aquatic habitats is green algae. However, primitive plant species like bryophytes, which have poorly developed tissues, thin and incomplete cuticle layers and no root system [3], are more permeable and completely absorb all of the required nutrients and minerals including other elements present in the environment. These major characteristic features make mosses more prominent and suitable as plants for the study. Due to their high affinity for binding ions even under unfavorable living conditions, they can be used as indicators for monitoring pollutants [4,5,6,7,8,9,10,11,12].

Water resources used for the purpose of irrigating agricultural lands must be free from these heavy metals, as crops growing in such contaminated water and soil can take up such pollutants through their roots. The uptake of these heavy metals, e.g., nickel, cadmium and lead, by the plants is very toxic even in trace amounts, and may be possibly consumed by humans, causing complex health conditions like cancer and different kinds of mutations. Heavy metals present in the surroundings in any form can enter the food chain irrespective of the level of the living organisms involved and can be highly harmful. Therefore, it is very important to keep monitoring the level of these contaminations in our ecosystem [13,14,15,16].

Mosses belong to the bryophytes group and occupy most parts of the Earth’s surface with a wide range of species. The flourishing of specific moss species varies in terms of morphology, which ultimately supports their growth under different and extreme climatic conditions, i.e., drought, salinity, flooding and high irradiation.

In order to monitor for the presence of heavy metals in aquatic habitats, moss species are selected based on various parameters, most importantly their morphological features and growth rate. The leaf size and total surface area of the leaf are directly related to the amount of uptake/absorption of heavy metals and binding of ions to the leaf surface, as the more leaf crowding, the higher the efficiency. Simultaneously, the slow rate of growth of the species helps in binding greater quantities of ions, producing several layers of deposition on the leaf surface [17,18,19]. Mosses have comparatively higher tolerance to pollutants and, therefore, are used for monitoring air, water and heavy metal pollution [20,21,22].

The diversity of heavy metals absorbed by these moss species is completely dependent upon the affinity of the chemical composition of the cell wall and cell membrane of the moss species, as maximum percentages of the heavy metals are deposited in the cell wall and the plasma membrane [17,23,24]. Various studies have shown that mosses act as a promising source by using numerous mechanisms for biomonitoring of heavy metals in all types of habitats [25,26,27,28].

The pH of the different moss species is important, as it reveals a lot about the type of metal binding possibility depending upon the functional groups present on the moss cell surface. The organic composition of the cell wall supports the formation of complex compounds with heavy metals due to the presence of the most dominant functional groups, such as carboxyl and phosphoryl groups [17]. It is no longer doubtful that the presence of a high percentage of heavy metals can have a negative effect on all of the catabolic and anabolic reactions in a wide range of plants, such as protein synthesis [29], photosynthesis, respiration, mineral uptake and membrane stability [30]. Many studies of the bryophytes show that abiotic stress, specifically heavy metal stress, can have major effects on the status of membrane permeability, leading to unbalanced ion leakage and an overall decrease in the chlorophyll concentration [31].

Bryophytes have been shown to possess great ion exchange capacity and stability of metal–organic complexes on their cell surfaces, like lichens [30]. The unique morphological features, such as unistratose phylloids and uniseriate rhizoids, of different moss species provide them with increased surface area and more volume for high rates of cation exchange. Therefore, mosses are assumed to have high tolerance to heavy metal stress as compared to other taxa [32].

In our study, we focused mainly on the accumulation of copper in the moss *Taxiphyllum barbieri* (Java moss). Copper is one of the most important metals involved in the activation of various enzyme systems, but most importantly, as an electron carrier in photosynthesis [33,34,35]. In both cases, either the deficiency or excessive presence of copper influences growth and viability [35]. In conditions when the concentration of Cu accumulated in plant tissue rises above the range tolerated by a healthy plant, i.e., 2–5 to 30 mg kg^−1^ dry weight (DW) [33], it becomes toxic. The toxicity due to Cu has various impacts on the plant; mainly, there is a decrease in the rate of photosynthesis and a reduction in the efficiency of photosystem II. The extent of toxicity varies depending upon the length of exposure to the Cu concentration and the genotype of the species. The moss *Taxiphyllum barbieri* (Java moss) is a common aquatic plant in habitats from Southeast Asia. The main aim of the study was to evaluate the sensitivity of *Taxiphyllum barbieri* to a high accumulation of Cu by analyzing the various physiological parameters. We observed that our selected moss is quite tolerant to copper, and, therefore, it can be helpful to study pollution control using this moss species.

## 2. Results

### 2.1. Cu and K Content

The accumulation of Cu under laboratory treatment of the moss *T. barbieri* increased with increased Cu concentrations in water. The average total Cu concentration in *T. barbieri* reached approximately 500 µg/g DW at 50 µM Cu concentration tested during 24 h of Cu exposure, while at 250 µM Cu concentration, it reached more than 1100 µg/g DW. At the highest level of effect, exposure to 500 µM Cu concentration yielded an average total Cu concentration in Java moss plants of approximately 3200 µg/g DW (Figure 1a).

Intracellular Cu concentrations were lower and reached approximately 300 µg/g DW at 50 µM Cu concentration tested during 24 h, or even up to 1700 µg/g DW (Figure 1b) at the highest external Cu dose tested (500 µM). The increase in intercellular Cu content in *T. barbieri* was statistically significantly negatively correlated with a decrease in intracellular K content, from approximately 20 mg/g DW (control) to approximately 4 mg/g DW at 500 µM Cu treatment (Figure 1c).

### 2.2. Composition of Assimilation Pigments and Chlorophyll a Fluorescence

After 24 h exposure to external Cu doses, we observed a significant decrease in chlorophyll *a* content from 50 µM Cu concentration tested (Figure 1d), while chlorophyll *b* content in moss was relatively stable (Figure 1e). Chlorophyll *a + b* (Figure 1f), similarly to the chlorophyll *a*/*b* ratio (Figure 1g), decreased as a result of 50 µM and higher Cu doses tested. Total carotenoid content in *T. barbieri* was sensitive to the presence of external Cu and statistically significantly decreased from 50 µM Cu doses tested (Figure 1h).

Chlorophyll *a* integrity after 24 h of Cu exposure significantly decreased at 50 µM and higher Cu doses as compared to the control (Figure 2a); fluorescence of chlorophyll *a* likewise decreased significantly, measured as F_V_/F_M_ (Figure 2b).

### 2.3. Soluble Proteins and TBARS

The concentration of soluble proteins in treatment as compared to control significantly decreased following external 500 µM Cu doses tested after 24 h of exposure (Figure 2c).

Thiobarbituric acid (TBARS) reactive substance production in *T. barbieri* was a parameter relatively sensitive to excess Cu and statistically significantly increased at 250 µM and higher external Cu doses (Figure 2d).

### 2.4. Content of Antioxidants

Content of reduced glutathione (GSH) in the moss *T. barbieri* significantly decreased at levels compared to control (Figure 2e). This phenomenon was followed by increased oxidized glutathione (GSSG) content, significant at 500 µM Cu treatment (Figure 2f). Due to this, ratios between GSH/GSSG (Figure 2g) were very sensitive parameters of Cu toxicity in the moss *T. barbieri*.

Ascorbic acid (AsA) content in *T. barbieri* (Figure 2h) significantly decreased at 50 µM Cu and was in fact not even detectable in any samples treated by external Cu doses.

### 2.5. Content of Selected Amino Acids and Related Compounds, Determination of Phenolic Acids

We observed an increase in 5-methylthioadenosine (MTA) content in *T. barbieri* due to excess Cu (Table 1). At the highest tested dose of Cu (500 µM), this increase was strong enough to be statistically significant. Significant increases in S-adenosylhomocysteine (SAH) and proline (Pro) contents in *T. barbieri* were also observed as a result of the highest tested dose of Cu when compared to controls (Table 1). On the other hand, we observed a significant Cu dose-dependent decrease in asparagine (Asn), cysteine (Cys), cystathionine and glutamate (Glu) in *T. barbieri* after 24 h of prolonged Cu exposure (Table 1). Contents of betaine, methionine (Met), serine (Ser) and taurine were not significantly affected by an excess of Cu in *T. barbieri* in our experiments (Table 1).

Significant increases in 3,4diOH benzaldehyde, chlorogenic acid, p-coumaric acid and especially cinnamic acid contents were observed as result of increased Cu content in *T. barbieri* (Table 1). On the other hand, contents of caffeic acid, sinapic acid, syringic acid, neochlorogenic acid, pOH benzaldehyde, pOH benzoic acid, protocatechuic acid, salicylic acid, vanillic acid and vanillin were relatively stable through all Cu treatments tested in this study (Table 1).

## 3. Discussion

### 3.1. Cu and K Content

It has been demonstrated previously that cryptogams (e.g., algae, mosses and lichens), without protective cuticles when compared to higher plants can accumulate Cu from their environment when exposed to excess Cu [36,37]. Cu contents in the moss plants determined in the present study, both total and intracellular, correlated with the availability of Cu in the aquarium water provided. The ratio between total and intracellular Cu was found to be very low in our study, from 1.67 at 50 µM external Cu dose tested to about 1.88 at 500 µM Cu dose tested. This shows that there was a significant amount of intracellular Cu accumulation from the aquarium water.

Potassium is a very mobile element that leaks out of the cell when plasma membrane permeability is altered in some way, which is why it has been used as an indicator of changes in membrane permeability [37]. Potassium, which is an important cell electrolyte, leaks out of the cell membranes when the membranes lose selective permeability due to metal doses. It is assumed that the degree of potassium leakage is proportional to the duration of stress response or concentration of xenobiotics in the environment [37]. Our results showed that the previously mentioned phenomenon was also true for our study. It showed statistically that the intercellular Cu content was in significantly negative correlation with intracellular potassium content.

### 3.2. Composition of Assimilation Pigments and Chlorophyll a Fluorescence

Photosynthesis and respiration of plants are sensitive to metal stresses. Chlorophyll is quite sensitive to excess Cu concentrations, which in turn makes photosynthesis sensitive to higher Cu content. Changes in the assimilation pigment composition of mosses are routinely used for assessment of stress [38]. Significant decreases in chlorophyll *a* were observed in *T. barbieri*. Cu in the Java moss significantly decreased its content from the lowest concentration tested. This result was unsurprising due to the many studies that have shown decreased chlorophyll *a* content to be a common sign of heavy-metal toxicity [39,40,41]. Chlorophyll *b* in the moss plants remained relatively stable throughout exposure to all of the Cu concentrations tested in this study.

At the lowest concentration (Cu dose 50 µM), total chlorophyll (*a + b*) decreased significantly in *T. barbieri*. There was also a significant decrease in the ratio of chlorophyll *a* to *b* (*a*/*b*) with increased doses of Cu. This is in accordance with the results observed previously for algae and lichens [39,42,43,44,45].

Total carotenoid content in *T. barbieri* was sensitive to the presence of external Cu and statistically significantly decreased from 50 µM Cu doses tested. This is in accordance with previously published studies, where authors demonstrated that total content of carotenoids was sensitive to the presence of some heavy metals (e.g., copper) in short-term experiments [39]. The phaeophytinization quotient (chlorophyll *a* integrity) reflects the ratio of the degradation of chlorophyll *a* to phaeophytin *a* [46]. In healthy lichens, mosses and algae, the OD435/OD415 ratio is about 1.4 [47], which is in accordance with the results obtained for untreated moss plants in this study. In the presence of Cu, a marked decrease in this value was observed in two lichens [48,49], which agrees with results obtained in the present study. Chlorophyll *a* fluorescence is another useful technique that allows us to obtain rapid information on the photosynthesis of mosses [38]. Lower values, as found in this study, indicate damage to Photosystem II (PSII) and hence a decreased photosynthetic efficiency. The toxicity of Cu to PSII was also confirmed in the case of lichens and algae [36,37].

### 3.3. Content of Soluble Proteins and TBARS

Soluble proteins exhibit a marked scavenging effect on reactive oxygen species (ROS), including superoxide and hydrogen peroxide in diverse plants. Although the amounts of soluble proteins in mosses are relatively easily determinable [50], their content is still not very frequently used as a parameter for the assessment of environmental stress, including that from heavy metals. There are a variety of products of the damage produced by oxidative stress, one of them being TBARS. The TBARS assays measure the lipid peroxidation and produce malondialdehyde (MDA), a reactive aldehyde produced by lipid peroxidation of polyunsaturated fatty acids. MDA has been found to be positively correlated with the degree of membrane lipid peroxidation and increased ion permeability [51]. This process is sensitive to the excess of some heavy metals, including Cu. In our study, we observed a decrease in soluble protein contents in *T. barbieri*, positively correlated with increased Cu doses in the environment. These data are in accordance with previously published studies for cryptogams [39,52,53,54].

### 3.4. Role of Antioxidants in Cu Tolerance/Toxicity

Redox-active metals, including Cu, can lead to the production of reactive oxygen species (ROS) in living cells. Oxidation may result in the formation of O^2−^ and subsequently H_2_O_2_ and a hydroxyl radical via Fenton-type reactions [55]. Several reports showed that elevated levels of glutathione (GSH) and ascorbate (AsA) are associated with increased oxidative stress tolerance [56,57,58]. Our short 24 h study showed that both GSH and AsA levels significantly decreased from the lowest 50 µM Cu doses tested, indicating that the moss was not tolerant of the excess Cu. We expected that these antioxidants would be used as part of complex defense mechanisms against ROS production caused by redox-active metals such as copper. The glutathione-ascorbate cycle is one of the major antioxidant mechanisms in cells. Ascorbate peroxidase (APX) and glutathione reductase (GR) activities were shown to increase in response to different types of stress exposure in plants, and it was confirmed by analyzing the mosses [59,60] growing near metal-polluted industrial areas. We did not find synthesis of phytochelatins (PCs), heavy-metal binding peptides, in *T. barbieri* plants. Interestingly, while PCs were discovered in higher plants, lichens and green algae [39], we did not detect their synthesis in Java moss. It seems that the role of the PCs in metal homeostasis in mosses is still not sufficiently understood [61]. In the moss *Leptodictyum riparium*, increased activation of glutathione transferase and phytochelatin synthase as a response to cadmium stress was observed, while similarly to the results of our study, those authors did not observe an important role of phytochelatins in Cd phytoremediation [62].

### 3.5. Role of Amino Acids and Related Compounds, Determination of Phenolic Acids in Cu Tolerance/Toxicity

Abiotic as well as biotic stresses may cause significant changes in the methionine (MTC) enzymes [63]. The methylated macromolecules methionine, cysteine, homocysteine, and taurine are the four common sulfur-containing amino acids; however, only methionine and cysteine are incorporated into proteins. Sulphur-containing amino acids are very important for metal detoxification and homeostasis in plants, including mosses [39,64].

In the present study, methionine content was not significantly altered due to excess Cu exposure, while S-adenosylhomocysteine (SAH) content and 5-methylthioadenosine (MTA) content increased. It seems that 5′-Methylthioadenosine/S-adenosylhomocysteine (MTA/SAH) nucleosidase (MTAN) plays a key role in the methionine-recycling pathway of mosses, similarly to that in bacteria and higher plants [65]. Proline content was previously demonstrated as an important factor reflecting heavy metal excess, including excess Cu in algae, lichens and mosses [39]. Proline is predominantly synthetized from glutamate and ornithine. Increased content of proline in *T. barbieri* was negatively correlated with decreased glutamate content, and we assume that glutamate served as a substrate for proline. Cysteine is required for methionine and glutathione synthesis. Cysteine is unique among amino acids, as it contains a thiol (-SH) group, which is extremely important for metal homeostasis in plants, as it can bind heavy metals. Cysteine molecules are incorporated into a wide range of peptides and proteins that may chelate heavy metals in living cells [66]. In the present study, we also observed a decrease in cystathione content, which is important in cysteine synthesis. Asparagine was found to be able to bind to some metals, e.g., cadmium, lead and zinc [67,68]. It is possible that decreased Asn in *T. barbieri* can be explained also by the production of Asn complexes with Cu cations.

Phenolic acids and flavonoids function as reducing agents, free radical scavengers, and quenchers of singlet oxygen formation. The antioxidant activity of phenolic acids in metal homeostasis in mosses is far from being understood. In *T. barbieri*, we found that control plants were most abundant in phenolic acids such as pOH benzaldehyde, cinnamic acid, sinapic acid, syringic acid and vanillin. Excess Cu did not decrease the content of any of the tested phenolic acids in the short-term study, while contents of 3,4diOH benzaldehyde, chlorogenic acid, p-Coumaric acid and especially cinnamic acid were strongly increased as a result of excess Cu in *T. barbieri*. Previously observed results indicated that during heavy metal stress, phenolic compounds can act as metal chelators and, on the other hand, phenolics can directly scavenge molecular species of active oxygen [41,69].

## 4. Materials and Methods

### 4.1. Sample Cultivation, Collection and Cu Treatment

The aquatic moss *Taxiphyllum barbieri* (Cardot and Copp.) Z. Iwats (Bryophyta, Hypnaceae), native to Southeast Asia, was selected for this study (Figure 3). Plants were sampled from those cultivated in aquaria of the Botanical Garden of Pavol Jozef Šafárik University in Košice at approximately 23 °C under a 16 h photoperiod (30 µmol.m^−2^ s^−1^; “cool white” tubes).

After collection, samples were transferred in glass beakers (capacity of 0.5 L, containing tap water) to the laboratory. Macroscopic foreign material adhering to moss surfaces was removed with forceps.

Samples (approximately 100 mg FW of moss plants) were washed with deionized water and further submerged in natural aquarium water as control. For treatment, moss plants were placed in water enriched by 50, 250 and 500 µM Cu concentrations, in plastic tubes, with three replicates for each concentration with a total volume of 50 mL (pH 6.5). The tubes were placed for 24 h in a climatic chamber at 23 °C under a 16 h photoperiod using 30 µmol m^−2^ s^−1^ PPFD, cool white, fluorescent lights. Cu was supplied in the divalent form as CuCl_2_. The dry mass of the moss was determined by weighing sub-samples dried in an oven overnight at 90 °C.

### 4.2. Determination of Cu and K Content

For analyses of total Cu, intracellular Cu and intracellular K content, thalli of control mosses and mosses treated for 24 h with Cu in aquarium water were removed and subsequently rinsed with 10 mL of deionized water (analyses of total Cu content). Another set of identically treated moss plants were washed for 20 min in 20 mL of 10 mM Na_2_-EDTA (analytical grade) to remove nonspecifically bound Cu, and then rinsed with 10 mL of deionized water (analyses of intracellular Cu content and K content).

Moss plants were dried at 90 °C for 24 h and digested for 48 h in 3 mL of 65% HNO_3_ (Suprapur, Merck, Darmstadt, Germany) and 30% H_2_O_2_ (2:1, *v*/*v*), with the volume brought to 10 mL with deionized water, *n* = 3 [70]. Cu and K concentrations in the samples were then determined by flame atomic absorption spectroscopy (FAAS). Analysis was performed using a Perkin-Elmer 3030B spectrometer (Perkin-Elmer Corp., Norwalk, CT, USA), detection limits: Cu (1.5 µg/L), K (1.5 µg/L), RSD = 6.9%. Each sample was analyzed at least three times, and mean values were used as one observation. Three replicates for each treatment in both sets of experiments were analyzed. Quality assurance/quality control was provided by using ACS or better-grade chemicals, analysis of all reagent blanks and calibration with standards. Standard N9300224 (Perkin-Elmer, Waltham, MA, USA, 2021) was used for initial calibration verification (ICV).

### 4.3. Determination of Assimilation Pigments

For analyses of composition of assimilation pigments, moss subsamples (approximately 15 mg DW) treated for 24 h with Cu were extracted in the dark for 1 h at 65 °C in 5 mL of dimethyl sulfoxide (DMSO). Extracts were allowed to cool to ambient temperature, and the absorbance, a reflection of turbidity, was checked at 750 nm with a UVI Light XTD 2 spectrophotometer (Secomam, Alès, France) to be certain that it was always less than 0.01.

To assess the amount of chlorophyll, the absorbance of extracts was read at 665.1, 649.1, 435 and 415 nm on the spectrophotometer [71]. Absorbance was also read at 480 nm to assess total carotenoids. Chlorophyll *a*, chlorophyll *b*, chlorophyll *a + b* and total carotenoids were calculated using equations derived from specific absorption coefficients for pure chlorophyll *a* and chlorophyll *b* in DMSO [71]. The ratio of optical densities at 435 and 415 nm (OD 435/OD 415), termed the phaeophytization quotient (or chlorophyll *a* integrity), was interpreted as reflecting the ratio of chlorophyll *a* to phaeophytin *a*. Three replicates were used.

### 4.4. Determination of Soluble Proteins and TBARS

For analysis of soluble proteins, moss thalli were homogenized in an ice-cold mortar in 50 mM phosphate buffer (pH 6.5). After centrifugation at 15,000× *g* at 4 °C for 20 min, the water-soluble protein content of supernatants was measured using the methods of Bradford [50]. Supernatants (30 μL) were pipetted into 970 μL of Bradford assay kit (Biorad, Hercules, CA, USA) in a spectrophotometric cuvette and mixed. After 10 min, absorbance of samples was spectrophotometrically measured at 595 nm. Bovine serum albumin was used as a calibration standard. Three replicates were used for each variant of the experiment.

The membrane lipid peroxidation state in thalli of Java moss was estimated using thiobarbituric acid reactive substances assay (TBARS) [36]. Moss sub-samples (in hydrated, metabolic active state) were homogenized in a mortar using ice-cold 10% (*w*/*v*) trichloroacetic acid (TCA) with addition of Whatman CF/C filters (glass-fiber filters that facilitate disruption of cell walls). The homogenate (2 mL final volume) was centrifuged at 10,000× *g* for 10 min. The supernatant (1 mL) was added to 1 mL of 0.6% thiobarbituric acid (TBA) in 10% TCA. After treatment of samples in an oven (99 °C) for 25 min and immediate cooling of tubes in an ice bath, the mixture was again centrifuged at 10,000× *g* for 10 min. Absorbance of the supernatant was measured at 532 nm (extinction coefficient for MDA–TBA complex 155 mM^−1^ cm^−1^) and corrected for non-specific absorption at 600 nm. Three replicates were used for each time and variant of the experiment.

### 4.5. Determination of Glutathione (GSH, GSSG), Ascorbic Acid (AsA), Selected Amino Acids and Related Compounds as Well as Determination of Phenolic Acids

Reduced GSH, oxidized glutathione (GSSG) and ascorbic acid (AsA) were extracted from moss samples with 0.1 M HCl (0.1 g FW/1 mL) and quantified using LC-MS/MS (Agilent 1200 Series Rapid Resolution LC system coupled on-line to an MS detector Agilent 6460 Triple quadrupole with Agilent Jet Stream Technologies) at *m*/*z* values 308/76, 613/231 and 177/95 in positive MRM mode, respectively. Separation was performed using column Zorbax EC-C18 100 × 4.6 mm, 2.7 µm particle size and mobile phase consisting of 0.2% acetic acid and methanol (95:5). The flow-rate was 0.6 mL/min, and column temperature was set at 25 °C. Freshly prepared standards were used for calibration and quantification [72].

Liquid chromatography with tandem mass spectrometry using a triple-quadrupole MS detector was used to analyze secondary metabolites. A volume of 1 mL of 0.1 M HCl (for determination of methionine cycle substances) or 80% methanol for determination of phenolic compounds and glass beads 0.5 mm in diameter were added to 10 mg of lyophilized sample. Samples were homogenized at 6800 rcf, centrifuged at 16,000× *g* rcf and then filtered through Whatman Mini-UniPrep syringeless filters 0.45 μm before analysis by liquid chromatography with mass detection. The samples were analyzed using an Agilent 1200 Rapid Resolution LC system. The system was connected to an Agilent Technologies 6460 triple-quadrupole MS detector with an Agilent Jet Stream, all from Agilent Technologies, Waldbronn, Germany. This system is described in more detail in a previously published work [72,73].

### 4.6. Statistical Analysis

One-way analysis of variance and Tukey’s pairwise comparisons (MINITAB Release 11, 1996) i.e., control compared to treatment, were used to determine the significance (*p* < 0.05) of differences in all measured parameters.

## 5. Conclusions

We are still looking for a model plant to evaluate the sensitivity and tolerance of aquatic plants to environmental pollution. In the present work, the physiological response of the aquatic moss *Taxyphyllum barbieri* to excess Cu was tested in a short-term, 24 h study. Mosses, similarly to algae (conventionally used in aquatic pollution studies), may or may not contain a proper and continuous protective layer of cuticle on their leaf surfaces. We found that excess Cu in the water (cultivation medium) positively correlated with total, as well as intracellular, Cu content in moss plants. Intracellular Cu content negatively correlated with intracellular K content, probably as a result of leakage of electrolytes caused by excess Cu. Excess Cu caused a decrease in chlorophyll *a* content; however, chlorophyll *b* content was relatively stable throughout the 24 h experiment; chlorophyll *a + b*, similarly to the chlorophyll *a/b* ratio, decreased due to increased Cu content in the water. Total carotenoid content decreased with the increases in the Cu concentrations tested. An excess of Cu caused increased phaeophytization of chlorophyll *a*, which means that the integrity of chlorophyll *a* decreased. Soluble protein content decreased due to excess Cu in water, and excess Cu caused increased production of TBARS, which negatively correlated with intracellular K content, suggesting degradation of cell plasma membranes.

Excess Cu caused changes in the content of selected antioxidants and decreased the content of reduced glutathione (GSH), while the content of oxidized glutathione (GSSG) increased, strongly decreasing the GSH/GSSG ratio. The content of ascorbic acid (AsA) strongly decreased in response to excess Cu from the lowest tested Cu concentration of 50 µM. Contents of amino acids and related compounds, as well as phenolic acid, were affected by excess Cu in the medium.

Based on the results of the present work, we assume that Java moss, *T. barbieri*, can be used as a model organism for future studies assessing the sensitivity of aquatic organisms to different kinds of xenobiotics in the environment.

## Figures and Tables

**Figure 1 plants-12-03607-f001:**
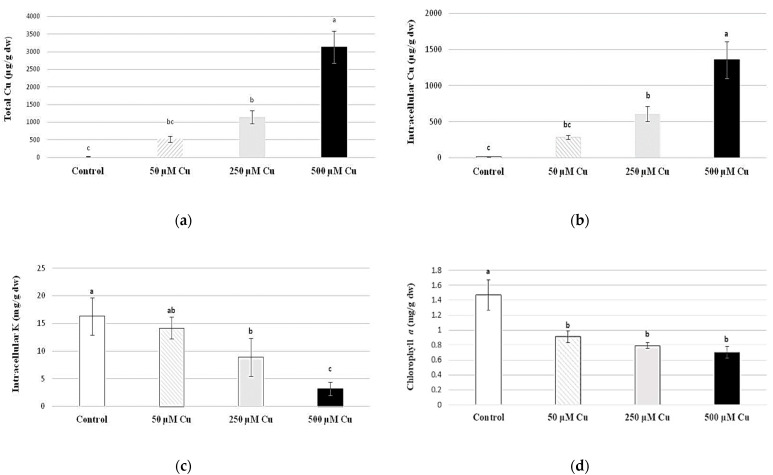
Total and intracellular Cu contents (**a**,**b**, µg/g dw), intracellular K content (**c**, mg/g dw), chlorophyll *a*, chlorophyll *b* and chlorophyll *a* + *b* contents (**d**–**f**, mg/g dw), chlorophyll *a*/*b* ratio (**g**) and content of total carotenoids (**h**, mg/g dw) in moss *Taxiphyllum barbieri* treated for 24 h with selected Cu concentrations (0, 50, 250 and 500 μM). Data are means ± SDs (*n* = 3). Values in vertical lines followed by the same letter(s) are not significantly different according to Tukey’s test (*p* < 0.05).

**Figure 2 plants-12-03607-f002:**
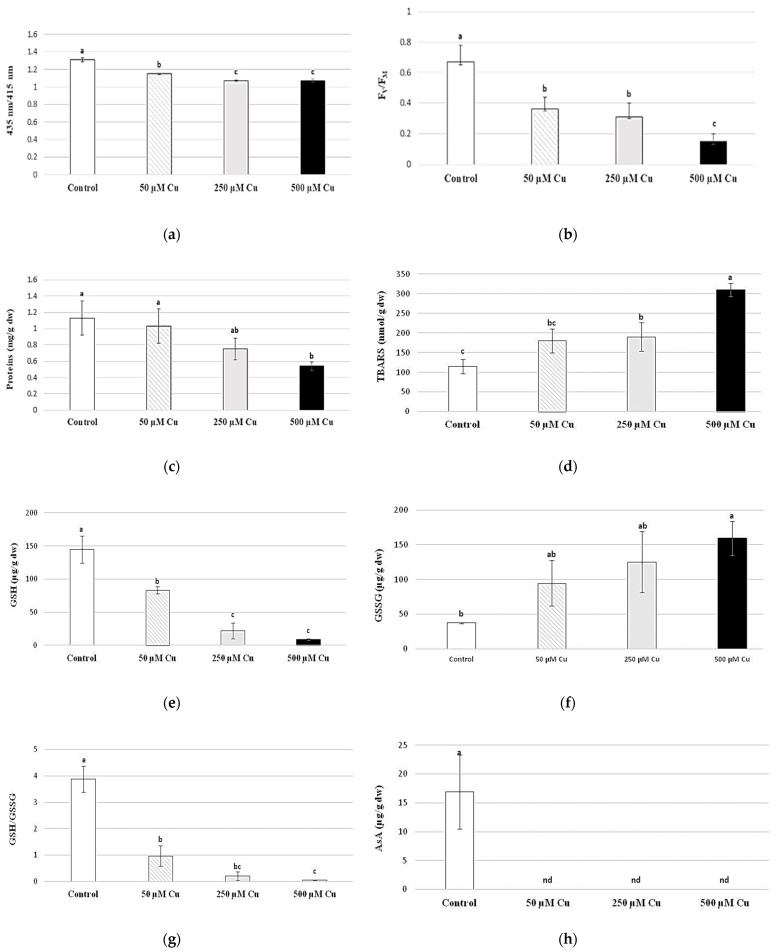
Chlorophyll a integrity (**a**, A 435 nm/415 nm), chlorophyll a fluorescence (**b**, F_V_/F_M_), soluble protein content (**c,** mg/g dw), TBARS (**d**, nmol/g dw), GSH and GSSG (**e**,**f**, µg/g dw), GSH/GSSG ratio (**g**) and AsA content (**h**, µg/g dw) in *Taxiphyllum barbieri* treated for 24 h with selected Cu concentrations (0, 50, 250 and 500 μM). Data are means ± SDs (*n* = 3). Values in vertical lines followed by the same letter(s) are not significantly different according to Tukey’s test (*p* < 0.05).

**Figure 3 plants-12-03607-f003:**
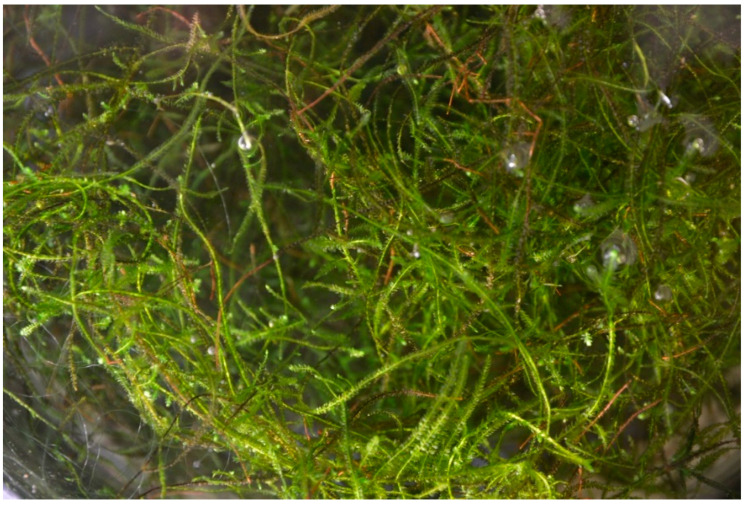
*Taxiphyllum barbieri* in aquarium.

**Table 1 plants-12-03607-t001:** Contents of 5-methylthioadenosine (MTA), S-adenosylhomocysteine (SAH), asparagine (Asn), betaine, cysteine (Cys), cystathionine, glutamate (Glu), methionine (Met), proline (Pro), serine (Ser), taurine, 3,4diOH benzaldehyde, caffeic acid, chlorogenic acid, sinapic acid, syringic acid, neochlorogenic acid, p-coumaric acid, pOH benzaldehyde, pOH benzoic acid, protocatechuic acid, salicylic acid, vanillic acid, vanillin and cinnamic acid in *Taxiphyllum barbieri* treated for 24 h with selected Cu concentrations (0, 50, 250 and 500 μM). Data are means ± SDs (*n* = 3). Values in vertical lines followed by the same letter(s) are not significantly different according to Tukey’s test (*p* < 0.05).

µg/g DW	Control	50 µM Cu	250 µM Cu	500 µM Cu
MTA	2.49 ± 0.8b	4.01 ± 0.94b	3.65 ± 1.28b	8.43 ± 1.29a
SAH	1.26 ± 0.09b	2.05 ± 0.39ab	1.87 ± 1.0b	3.57 ± 0.45a
Asn	151 ± 46.1a	96.5 ± 28.9ab	35.3 ± 42.4b	32.5 ± 21.2b
Betaine	27.7 ± 12.1a	32.7 ± 17.3a	14.3 ± 8.53a	15.9 ± 6.31a
Cys	2.13 ± 0.16a	0.10 ± 0.05b	0.06 ± 0.04b	0.04 ± 0.01b
Cystathionine	0.11 ± 0.02ab	0.13 ± 0.02a	0.07 ± 0.02bc	0.06 ± 0.01c
Glu	98.4 ± 12.1a	57.6 ± 12.4b	75.6 ± 13.6ab	51.4 ± 7.3b
Met	10.4 ± 1.6a	10.72 ± 2.76a	7.9 ± 3.7a	7.6 ± 1.1a
Pro	50.9 ± 11.9b	56.8 ± 8.6b	58.6 ± 9.8b	85.4 ± 8.5a
Ser	282 ± 109a	312 ± 56a	268 ± 40.0a	331 ± 82.1a
Taurine	1.75 ± 0.09a	1.85 ± 0.83a	2.81 ± 0.47a	2.71 ± 1.84a
3,4diOH Benzaldehyde	48.8 ± 34.3b	449 ± 219a	327 ± 132a	902 ± 183a
Caffeic acid	44.8 ± 24.8a	55.2 ± 52.7a	56.3 ± 4.43a	82.5 ± 52.8a
Chlorogenic acid	425 ± 264b	717 ± 30.5ab	681 ± 65.6ab	960 ± 160a
Sinapic acid	2324 ± 796a	1612 ± 187a	1450 ± 486a	1197 ± 202a
Syringic acid	1493 ± 854a	1070 ± 523a	815 ± 280a	591 ± 178a
Neochlorogenic acid	127 ± 31.4a	198 ± 65.3a	176 ± 31.2a	218 ± 65.7a
p-Coumaric acid	465 ± 72.7b	3094 ± 1833ab	2424 ± 1725ab	4994 ± 296a
pOH Benzaldehyde	5613 ± 1008a	7203 ± 2095a	4893 ± 738a	5567 ± 1295a
pOH Benzoic acid	282 ± 60.1a	155 ± 71.3a	127 ± 51.5a	176 ± 72.2a
Protocatechuic acid	60.1 ± 18.2a	82.6 ± 7.93a	113 ± 48.3a	92.9 ± 12.0a
Salicylic acid	188 ± 33.0a	171 ± 40.6a	43.2 ± 33.3b	158 ± 50.0a
Vanillic acid	411 ± 72.5ab	350 ± 84.2b	358 ± 31.4b	560 ± 87.6a
Vanillin	973 ± 244a	977 ± 62.2a	569 ± 100b	788 ± 138ab
Cinnamic acid	5395 ± 472b	16,664 ± 6817ab	10,507 ± 5189b	22,272 ± 2552a

## Data Availability

The data can be provided by the authors upon reasonable request.

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
