# Peer review of "Mechanisms of Copper Toxicity and Tolerance in the Aquatic Moss Taxiphyllum barbieri"

_plants, 2023, doi:10.3390/plants12203607_

Round 1

Reviewer 1 Report

There are many appropriate references on bryophytes that could be used instead of or in addition to those on algae and lichens.

I would like to see some statement or paragraph in the introduction to justify why you chose to look at the particular biochemical parameters you chose.

This paper needs a lot of English editing, which I have noted on the uploaded manuscript.  The Reference setion is inconsistent and does not follow the style of the journal.  Some places I could not understand what the authors were trying to say, so I was unable to correct them.

The research itself seems sound, but some critical information is missing in the methods, so I don't know if they are valid.  In particular, I don't know how the measurements were replicated.  Were all three replicates kept in the same flask?  Were they randomly located?  Did they represent different original aquaria?  I suspec several levels of pseudoreplication.  That should at least be acknowledged.

Author Response

Questions from Review 1
Q. 1. Were replicates in different flasks? Answer Yes, all the replicated were in different flasks.
Q. 2. Did they come from different aquaria or all from one? Answer All moss material was grown in one great, 500 L aquarium.
Q. 3. Were replicates arranged randomly to avoid position effect? Answer Yes, it is routine procedure used in our laboratory.
Q. 4. How many replicates did you use per treatment? Answer Three replicates each.
Q. 5. Was each replicate in a separate container? Answer Yes.
Q. 6. What physiological analyses? Answer Determination of assimilation pigments, soluble proteins, TBARS, GSH, GSSG, AsA, amino acids and phenolic acids (all measurements in the experiments, control, as well as Cu treated samples).
Q. 7. Did you determine the field concentration of Cu? Answer Yes, Cu content was lower than 3 μg/L in original, Cu untreated aquaria.
Q. 8. How did you avoid measuring materials on the surface? Did you wash the moss? If so, what did you use? Answer Samples were quickly rinsed with deionized water.
Q. 9. I'm confused. I thought they were taken from aquaria. What is natural water?
If not from aquaria, describe the original habitat. Answer “Natural aquarium water” means water which is regularly used for filling of aquaria in Botanical garden.

Reviewer 2 Report

The paper „ Mechanisms of Copper Toxicity and Tolerance in Aquatic Moss Taxiphyllum barbierifits the scope of the journal. Monitoring of water pollution is currently one of the major issues studied in numerous scientific centers due to the high heavy metal sources and their toxicity to living organisms. However, I could not find the sufficient novelty of this paper.

There are also some points that the author needs to answer or modify.

1.     Introduction and Discussion: The introduction needs significant revision. Of the 62 literature references, only 7 are works from the last 10 years. Most of the literature from before 2000 is cited. In recent years, many papers have been published that deal with issues similar to those presented in this publication. Moreover, part of the work concerns the Aquatic Moss described here. I see no reason why the authors omit more recent data.

2.     Methodology: What guided the authors when choosing the copper concentration range of 50-500 mM?

3.     Methodology: The plant samples were dried at a high temperature of 90oC. It is very likely that at this temperature some of the plant components will evaporate or partially degrade (heat labile compounds). This may affect the test results, e.g. no effect of copper addition on the content of some of the tested organic compounds. How did the authors check for such undesirable effects?

4.     Methodology: Is decomposition of the plant material complete under the given conditions (did the plants contain no silica)?

5.     Lines 98,102 – please correct “Cu”.

6.     Lines 120 – please correct “K”.

7.     Lines 243, 247 – please give abbreviations “Pro”, “Met”, “Ser”.

8.     Lines 291-292 - if "many studies", then why is there only one reference (self-citation, by the way)?

9.     Line 298 - is there only work on this topic?

10.  Lines 329, 332 – please explain abbreviations: “ROS”, “AsA”.

11.  Line 330 – please correct O2-, H­2O2.

12.  Lines 377-380 - please consistently write chemical names in lowercase, like 249-253 and 256-259

13.  Literature - not in line with the journal's requirements

Author Response

Questions from Review 2
There are also some points that the author needs to answer or modify.
Q. 1. Introduction and Discussion: The introduction needs significant revision. Of the 62 literature
references, only 7 are works from the last 10 years. Most of the literature from before 2000 is cited. In
recent years, many papers have been published that deal with issues similar to those presented in this
publication. Moreover, part of the work concerns the Aquatic Moss described here. I see no reason
wh y the authors omit more recent data.
Answer we have added some more citations in the updated version of the manuscript.
Q
Q. . 2. Methodology: What guided the authors when choosing the copper concentration range of 502. Methodology: What guided the authors when choosing the copper concentration range of 50--500 mM?500 mM? AnswerAnswer--These These Cu Cu concentrations are frequently used in our laboratory for short toxicity tests concentrations are frequently used in our laboratory for short toxicity tests inin aquatic plants.aquatic plants.
Q.
Q. 3. Methodology: The plant samples were dried at a high temperature of 90oC. It is very likely 3. Methodology: The plant samples were dried at a high temperature of 90oC. It is very likely that at this temperature some of the plant components will evaporate or partially degrade (heat that at this temperature some of the plant components will evaporate or partially degrade (heat labile compounds). This may affect the test results, e.g. no elabile compounds). This may affect the test results, e.g. no effect of copper addition on the content of ffect of copper addition on the content of some of the tested organic compounds. How did the authors check for such undesirable effects?some of the tested organic compounds. How did the authors check for such undesirable effects? AnswerAnswer--This temperature (90This temperature (90ooC) was used only for analyses of subC) was used only for analyses of sub--samples, for calculation of dry samples, for calculation of dry weight of moss thalli (determination of water content in fresh thalli). All measurements (e.g. weight of moss thalli (determination of water content in fresh thalli). All measurements (e.g. pigments, proteins, TBARS, GSH, GSSG, AsA ... ) were done in moss material with known fresh pigments, proteins, TBARS, GSH, GSSG, AsA ... ) were done in moss material with known fresh weight.weight.
Q.
Q. 4. Methodology: Is decomposition of the plant material complete under the given conditions (did 4. Methodology: Is decomposition of the plant material complete under the given conditions (did the plants contain no silica)?the plants contain no silica)? AnswerAnswer--Decomposition of the plant material was complete under the given conditions.Decomposition of the plant material was complete under the given conditions.

Round 2

Reviewer 2 Report

Dear authors,

Thank you for taking into account some of my comments. However, your answers to questions about the analysis methodology prove that you do not pay enough attention to the correctness of the analyses. Mosses contain silicon and it is not possible to remove this element using only nitric acid and hydrogen peroxide, without hydrofluoric acid. I assume that the undecomposed silicon compounds were filtered out by the authors before analyzing the copper. In turn, drying plants at 90°C may increase the weight loss due to the release of volatile substances and the gradual oxidation of carbon.

Author Response

Mosses contain silicon and it is not possible to remove this element using only nitric acid and hydrogen peroxide, without hydrofluoric acid.

I assume that the undecomposed silicon compounds were filtered out by the authors before analyzing the copper.

Yes, undecomposed silicon compounds were filtered out prior to analyzing the copper.

In turn, drying plants at 90°C may increase the weight loss due to the release of volatile substances and the gradual oxidation of carbon.

Yes, we used air dried moss material for Cu analyses. Drying of plants at 90 °C was used only for calculation of water content in samples, like for other biochemical analyses used in this study.

Thank you for your comments and help,

Martin Backor, corresponding author